# Improving Network Training on Resource-Constrained Devices via Habituation Normalization [note 1]

**DOI:** 10.3390/s22249940

**Published:** 2022-12-16

**Authors:** Huixia Lai, Lulu Zhang, Shi Zhang

**Affiliations:** 1The College of Computer and Cyber Security, Fujian Normal University, Fuzhou 350007, China; 2The Digit Fujian Internet-of-Things Laboratory of Environmental Monitoring, Fujian Normal University, Fuzhou 350007, China

**Keywords:** normalization, neural network training, resource-constrained device, habituation, EEG signal application

## Abstract

As a technique for accelerating and stabilizing training, the batch normalization (BN) is widely used in deep learning. However, BN cannot effectively estimate the mean and the variance of samples when training/fine-tuning with small batches of data on resource-constrained devices. It will lead to a decrease in the accuracy of the deep learning model. In the fruit fly olfactory system, the algorithm based on the “negative image” habituation model can filter redundant information and improve numerical stability. Inspired by the circuit mechanism, we propose a novel normalization method, the habituation normalization (HN). HN first eliminates the “negative image” obtained by habituation and then calculates the statistics for normalizing. It solves the problem of accuracy degradation of BN when the batch size is small. The experiment results show that HN can speed up neural network training and improve the model accuracy on vanilla LeNet-5, VGG16, and ResNet-50 in the Fashion MNIST and CIFAR10 datasets. Compared with four standard normalization methods, HN keeps stable and high accuracy in different batch sizes, which shows that HN has strong robustness. Finally, the applying HN to the deep learning-based EEG signal application system indicates that HN is suitable for the network fine-tuning and neural network applications under limited computing power and memory.

## 1. Introduction

At present, many applications based on neural networks are embedded in portable devices to monitor the IoT system in real-time. However, most of the portable devices are resource-constrained, such as limited power, limited computing power and limited memory space. Training/fine-tuning neural networks on resource-constrained devices often requires setting different training parameters from the original networks, which may lead to a decrease in accuracy and affect the final application. For example, when fine-tuning on embedded application systems, a smaller batch size often significantly affects the accuracy of the neural network. Through analysis, we find that the accuracy drop of fine-tuned neural networks is related to the sensitivity of normalization to the batch size.

Normalization can improve the training efficiency and generalization ability of neural network models. Therefore, normalization has been an influential component and an active research field of deep learning, promoting the development of some research fields, such as computer vision and machine learning. Among normalization methods, the batch normalization (BN) [1] normalizes by calculating the mean and variance within a batch of data before the activation. BN helps to stabilize the distribution of internal activations during model training. Numerous experiments show that BN can effectively improve the learning efficiency and the accuracy of deep learning networks [2]. BN is a foundation of many state-of-the-art computer vision algorithms and is applied to the latest network architectures.

BN, with great success, is not without drawbacks. For example, on the ResNet-50 model trained in CIFAR10, BN performs well with a sufficiently large batch size (e.g., 32 images per worker). However, a small batch size leads to an inaccurate estimation of the mean and variance within a batch, leading to reduced model accuracy (Figure 1). In addition, BN cannot be effectively applied to recurrent neural networks (RNNs) [3]. In response to this problem, some normalization methods have been proposed. For example, Layer Normalization (LN) [3], Weight Normalization (WN) [4], Instance Normalization (IN) [5], Group Normalization (GN) [6], Attentive Graph Normalization (AGN) [7], etc. GN has higher stability among these normalization methods, but lower performance in medium and large batches. As a particular case of BN and LN, IN only considers all elements of a channel in one sample to calculate the statistics, which is more suitable for fast stylization. LN is mainly applied to recurrent neural networks (RNNs) and is rarely used in CNN. Therefore, it is necessary to explore a new normalization method with high stability and suitability for different network types.

Habituation is a type of non-associative plasticity in which neural responses to repeated neutral stimuli are suppressed over time [8]. Habituation in biology is applied in robotics applications [9,10] and deep learning networks to enhance object recognition [11]. Habituated models are also applied to information filtering, pattern classification and anomaly detection to improve the anomaly detection accuracy [12]. These studies reveal the benefits of using habituation in machine learning and suggest that the models incorporating additional features of habituation could yield more robust algorithms. In this paper, we propose a habituation normalization method (HN) based on the habituation “negative image” model by calculating the suppressed image of the input data and then normalizing the input by subtracting the inhibitory image. HN uses batches of data to construct the inhibitory picture and achieves a batch size independent normalization method. It can also effectively eliminate noise or confusion in the statistical calculation. For example, training ResNet-50 on CIFAR10 with a batch size of 4, BN achieves the average test accuracy of the last five epochs of 56.58% while HN achieves 72.54% with notable improvement.

The main contributions of this paper are:We proposed a new normalization method, Habituation Normalization (HN), based upon the idea of habituation. HN can accelerate the convergence and accuracy of networks in a wide range of batch sizes.HN helps maintain the model’s stability. It avoids the accuracy degeneration when the batch size is small and the performance saturating when the size is significant.Experiments on LeNet-5, VGG16, and ResNet-50 show that HN has good adaptability.The application of HN to deep learning-based EEG signal application system shows that HN is suitable for deep neural networks running on resource-constrained devices.

In the remainder of this paper, we first introduce the works related to normalization and habituation in Section 2. Then the formulation and implementation are discussed in Section 3. In Section 4, the experimental analyses of HN are performed. Section 5 is a case study. Section 6 concludes the paper.

## 2. Related Works

### 2.1. Normalization

Normalizing hidden features in neural networks can speed up the network training, so normalization methods are widely used. In recent years, normalization methods, such as Batch Normalization (BN) [1], Layer Normalization (LN) [3], Weight Normalization (WN) [4], Instance Normalization (IN) [5], Batch Renormalization (BRN) [13], Group Normalization (GN) [6], and Attentive Graph Normalization (AGN) [7], are proposed successively. We briefly introduce these normalization methods in this subsection.

Ioffe and Szegedy proposed the batch normalization (BN) [1] in 2015. First, BN normalizes a feature map with the mean and variance calculated along with the batch, height, and width of the feature map. Then, BN re-scales and re-shifts the normalized feature map. It is widely used in CNN networks with significant results [14,15] but less applicable to RNN and LSTM networks. In addition, BN leads to the deterioration of network accuracy when the batch size is small.

In 2016, Ba, Kiros and Hinton proposed the layer normalization (LN) [3]. LN computes the mean and variance along a feature map’s channel, height, and width dimensions and then normalizes it. LN and BN are perpendicular to each other in terms of the dimensions where they find the mean and the variance. LN requires the same operation in the training and testing processes. It solves the problem that BN is unsuited for RNN, and at the same time, achieves good results when setting a small batch size. However, LN is still less accurate than BN in many large image recognition tasks.

Salimans and Kingma proposed the weight normalization (WN) [4] in 2016. WN decouples the weight vector into a parameter vector *v* and a parameter scalar *g* to reparametrize and optimize these parameters by stochastic gradient descent. Unlike BN and LN, WN has a special idea of parameter normalization. WN also accelerates the convergence of stochastic gradient descent optimization.

In 2016, Ulyanov, Vedaldi, and Lempitsky proposed the instance normalization (IN) [5]. IN takes all elements of a single sample, a single channel in a batch sample to calculate the mean and variance, and then normalizes. IN is mainly applied in style transfer to accelerate model convergence and maintain the independence between image instances.

In 2017, Ioffe proposed the batch renormalization (BRN) by adding two non-training parameters, *r* and *d*, to BN [13]. BRN keeps the equivalence of the training phase and the inference phase, and solves the problems of non-independent identical distribution and small batch. Although BRN solves the problem of the BN’s accuracy reduction in minor batch sizes, BRN is still batch dependent. Therefore, its accuracy is still affected by the batch size.

Wu and He proposed the group normalization (GN) [6] in 2018. GN divides the channel data into groups and calculates the mean and the variance of the channel, height, and width dimensions on each group. LN, IN, and GN all perform in dependent computations along the batch axis. The two extreme cases of GN are equivalent to LN and IN. Although GN is batch size independent, it needs to be divided into *G* groups. Therefore, its stability is between IN and LN.

Chen et al. proposed the attentive graph normalization (AGN) [7] in 2022. AGN learns a weighted combination of multiple graph-aware normalization methods, aiming to automatically select the optimal combination of multiple normalization methods for a specific task. However, it is limited to graph-based applications.

### 2.2. Biological Habituation and Applications

Habituation [8,9] is a form of simple physical memory. Over time, habituation inhibits the neural responses to repetitive, neutral stimuli, that is, behavioral responses will decrease when stimuli are perceived repeatedly. Habituation is also considered to be a fundamental mechanism of adaptive behavior, which is present in animals ranging from the sea slug Aplysia [16,17] to humans [18] through toads [19,20] and cats [21]. This adaptive mechanism allows organisms to focus their attention on the most salient signals in the environment, even when these signals are mixed with high background noise.

Some researchers [9,22] investigated the mechanism of short-term habituation in the fruit fly olfactory circuit and tried to reveal how habituation in early sensory processing affects the downstream occurrence of odor encoding and odor recognition. For example, a dog sitting in a garden that is habituated to the smell of flowers is likely to detect the appearance of a coyote in the distance, even though the odor of the coyote in the distance is only a tiny fraction of the original odor that enters the dog’s nose (Figure 2).

The effect of habituation on background elimination has also attracted the attention of computer scientists. Some computational methods that demonstrate the primary effects of habituation (i.e., background subtraction) have been used in robotics applications [9,10] and deep learning networks to enhance object recognition [11]. In 2018, Kim et al. applied the background subtraction algorithm [11] to each video frame, finding the region of interest (ROI). They then performed CNN classification and obtained the ROI as one of the predefined categories. In 2020, Shen et al. implemented an unsupervised neural algorithm for odor habituation in fruit fly olfactory circuits [9] and published the work in PNAS. They used background elimination to distinguish between similar odors and improved the prospect detection. The method improves the detection of novel components in odor mixtures.

Studies in [8,9,10,11,22] revealed the benefits of using habituation in machine learning or deep learning and suggested that models that incorporate additional features of habituation yielded more robust algorithms.

In this paper, based on the features that habituation can filter redundant information and make the values stable, we design a habituated normalized layer (HN) for neural networks. It can enhance the training efficiency of the network and improve the model accuracy.

## 3. Method

In this section, we first review existing normalization methods and then propose the HN method with stimulus memorability.

### 3.1. The Theory of Existing Normalization

The existing normalization methods calculate statistics from some dimensional ranges of the batch data and normalize. The objectives are unifying the magnitudes, speeding up the solution of gradient descent, avoiding neuron saturation, reducing gradient disappearance, preventing the small values in the output data from being swallowed and avoiding numerical problems caused by the large values. Take CNN as an example. Let *x* be the input data to an arbitrary normalization layer, represented as a 4-dimensional tensor [N,C,H,W], where *N* is the number of samples, *C* is the number of channels, *H* is the height, and *W* is the width. Let $x_{nchw}$ and ${\hat{x}}_{nchw}$ be pixel values before and after normalization, where n∈[1,N],c∈[1,C],h∈[1,H], and w∈[1,W]. Assuming that μ and σ are a mean and a standard deviation, respectively, the values normalized by BN, LN, and IN can all be expressed as (Equation 1).
(1)$${\hat{x}}_{nchw}=\;\alpha \frac{x_{nchw}-u}{\sqrt{\sigma^{2}+\epsilon }}+\beta$$
where α and β are a scale and a shift parameter, respectively, ϵ is a tiny constant.

Equation (Equation 1) summarizes the three normalizing calculation formulas of BN, LN, and IN. The only difference is that the pixels used to estimate μ and σ are different. μ and σ can be expressed using (Equation 2) and (Equation 3).
(2)$$u_{i}=\frac{1}{\left|S_{i}\right|}\sum_{\left(n,c,h,w\right)\in S_{i}}x_{nchw}$$
(3)$$\sigma_{i}^{2}=\frac{1}{\left|S_{i}\right|}\sum_{\left(n,c,h,w\right)\in S_{i}}{(x}_{nchw}-u_{i})$$
where i∈bn,ln,in is used to distinguish different methods, $S_{i}$ is a set of pixels, $\left|S_{i}\right|$ is the size of $S_{i}$.

BN counts all pixels on a single channel, which can be expressed as $S_{bn}=\left\{\left(n,h,w\right)|n\in \left[1,N\right],w\in \left[1,W\right],h\in \left[1,H\right]\right\}$. LN counts all pixels on a single sample, which can be expressed as $S_{ln}=\left\{\left(c,h,w\right)|c\in \left[1,C\right],w\in \left[1,W\right],h\in \left[1,H\right]\right\}$. IN counts all pixels of a single channel of a single sample, which can be expressed as $S_{in}=\left\{\left(h,w\right)|w\in \left[1,W\right],h\in \left[1,H\right]\right\}$.

For GN, *G* is the number of groups, which is a predefined hyperparameter (*G* = 32 by default). In GN, the size of the tensor is [N,G,C/G,H,W]. Let ${\;x}_{ncghw}$ and ${\hat{x}}_{ncghw}$ be pixel values before and after normalization, where $n\in \left[1,N\right],\;c\in \left[1,G\right],g\in \left[1,\left\lfloor \frac{C}{G}\right\rfloor \right],h\in \left[1,H\right],w\in \left[1,W\right]$. Then, Equations (Equation 1)–(Equation 3) are changed to (Equation 4)–(Equation 6).
(4)$${\hat{x}}_{ncghw}=\;\alpha \frac{x_{ncghw}-u_{gn}}{\sqrt{\sigma_{gn}^{2}+\epsilon }}+\beta$$
(5)$$u_{gn}=\frac{1}{\left|S_{gn}\right|}\sum_{\left(n,c,g,h,w\right)\in S_{gn}}x_{ncghw}$$
(6)$$\sigma_{gn}^{2}=\frac{1}{|S_{gn}|}\sum_{\left(n,c,g,h,w\right)\in S_{gn}}{(x}_{ncghw}-u_{gn})$$

GN divides the channels into several groups, and then counts all the pixels in each group, where $S_{gn}=\left\{\left(c,h,w\right)|c\in \left[1,\left\lfloor \frac{C}{G}\right\rfloor \right],h\in \left[1,H\right],w\in \left[1,W\right]\right\}$.

As can be seen from the above description, BN, LN, IN, and GN are all dependent on the data within a batch for calculating the mean and standard deviation. They do not consider the correlation between batches.

### 3.2. Habituation Normalization

Habituation has three general features in biology:Stimulus adaptation (reduced responsiveness to neutral stimuli with no learned or innate value).Stimulus specificity (habituation to one stimulus does not reduce responsiveness to another stimulus).Reversibility (de-habituation to context when it becomes relevant).

These features are closely relevant to computational problems, yet they have not been well applied.

Some researchers have established mathematical models of the habituation effects on the efficacy of a synapse, including Groves and Thompson [23], Stanley [24], and Wang and Hsu [25]. The model proposed by the Wang and Hsu considered a long-term memory, where the long-term memory means that an animal habituates more quickly to a stimulus to which it has been previously habituated. Shen et al. [9] developed an unsupervised algorithm (Figure 3). Inspired by habituation-related studies, we propose a novel habituation normalization method (HN) applicable to deep neural networks.

The “negative image” in HN is a weight vector *v*, which is initially a zero vector and has a shape of [1,C,1,1]. At iteration *t*, input $x_{nchw}$ is adjusted with (Equation 7).
(7)$$x_{nchw}=\;x_{nchw}-v_{nchw}^{t}$$
where the weight vector *v* is updated with (Equation 8), and the shape of *v* will match the shape of the input data.
(8)$$v_{nchw}^{t}=v_{1c11}^{t-1}+\gamma x_{nchw}-\varphi v_{1c11}^{t-1}$$

Then, the mean of *v* is calculated on the channel dimension, and the shape of *v* is adjusted to [1,C,1,1] to facilitate the following input (Equation 9).
(9)$$v_{1c11}^{t}=\frac{1}{\left|S\right|}\sum_{\left(n,c,h,w\right)\in S}v_{nchw}^{t}$$

Finally, the statistics and affine changes are calculated with (Equation 10)–(Equation 12).
(10)$$u_{HN}=\frac{1}{\left|S_{HN}\right|}\sum_{\left(n,c,h,w\right)\in S_{HN}}x_{nchw}$$
(11)$$\sigma_{HN}^{2}=\frac{1}{\left|S_{HN}\right|}\sum_{\left(n,c,h,w\right)\in S_{HN}}{(x}_{nchw}-u_{HN})$$
(12)$${\hat{x}}_{nchw}=\;\alpha \frac{x_{nchw}-u_{HN}}{\sqrt{\sigma_{HN}^{2}+\epsilon }}+\beta$$

In Equation (Equation 8), the habituation rate $\gamma \in \left(0,1\right)$, and the weight recovery rate $\varphi \in \left(0,1\right)$. In Equation (Equation 9), $S=\left(n,h,w\right)|n\in \left[1,N\right],w\in \left[1,W\right],h\in \left[1,H\right]$, |S|=N∗H∗W. In Equations (Equation 10) and (Equation 11), $S_{HN}=\left\{\left(n,c,h,w\right)|n\in \left[1,N\right],c\in \left[1,C\right],h\in \left[1,H\right],,w\in \left[1,W\right]\right\}$. In Equation (Equation 12), α and β are initialized as a one vector and a zero vector.

In the habituation method, the “negative images” are saved by vector *v*. If every batch of data is similar, we expect to form a “negative image” of the input with the time going on. After subtracting the “negative image” from the following input, what remains is the foreground components of the images. With HN, the construction of “negative image” is a gradual process. Therefore, the construction process considers the present batch data and the influence of the previous batch data simultaneously. Equation (Equation 9) removes the batch size factor after constructing a “negative image” process via (Equation 8). Equation (Equation 9) ensures that the HN is independent of the batch size. Therefore, it can be applied to different batch sizes.

### 3.3. Implementation

HN can be implemented by code in the popular neural network framework Pytorch. Figure 4 shows the code based on Pytorch.

## 4. Experiment

In this section, we evaluate the effectiveness of HN on two benchmark datasets and three different deep learning networks.

### 4.1. Experimental Setup

BN [1], GN [6], LN [3], BRN [13], and HN are utilized for comparison. We test them in three architectures for image classification: LeNet-5 [26], VGG16 [27], and ResNet-50 [28].

Two datasets are depicted in the following.
FASHION-MNIST [29]: FASHION-MNIST clones all the irrelevant features of the MNIST dataset: 60,000 training images and corresponding labels, 10,000 test images and related labels, 10 categories and 28 × 28 resolution per image. The difference is that FASHION-MNIST is no more extended abstract symbols but more concrete human necessities-clothing, with 10 types.CIFAR10 [30]: this dataset consists of 60,000 color images with 50,000 training images and 10,000 test images of 32 × 32 pixels, divided into 10 categories.

In the experiments, all deep learning models use cross-entropy loss, sigmoid as activation functions in convolutional neural networks, and ReLU as activation functions in residual networks. BN, LN, GN, BRN (https://github.com/ludvb/batchrenorm), and optimizer keep the default hyperparameters. As to HN, we set γ=0.5 , φ=0.1, t=4 as the default settings.

The experiments are performed on a machine with Intel(R) Xeon(R) Silver 4114 CPU 2.20 GHz 2.19 GHz (2 processors), 128 GB of RAM, NVIDIA Quadro P5000 graphics card, using Windows server 2016.

### 4.2. Comparisons on Convolutional Neural Networks

#### 4.2.1. LeNet-5

Following the idea of using simple networks by Ioffe and Szegedy [1], we build the vanilla convolutional neural network (Figure 5) according to the LeNet-5 structure proposed by LeCun [13]. The LeNet-5 consists of 2 convolutional layer blocks and 2 fully connected layer blocks. Each convolutional layer block includes a convolutional layer, a sigmoid activation function, and a maximum pooling layer. Each convolutional layer uses a 5 × 5 convolutional kernel. The first convolutional layer has 6 output channels, while the second one has 16 output channels. In the two maximum pooling layers, we set the kernel size to 2 × 2, stride to 2. To pass the output of the convolutional block to the fully-connected layer block, each sample is spread in a small batch. Three fully connected layers have 120, 84, and 10 outputs.

In the experiment, normalizations are inserted before sigmoid activation function. We did not apply any data enhancement methods to the FASHION-MNIST and CIFAR10 datasets. Each model was trained using Adam Optimizer with a learning rate of 0.001.

The first experiment was conducted on the FASHION-MNIST dataset. When batch size = 2, the classification accuracy of BN is much lower than that of vanilla CNN, LN, and HN (Figure 6a), which once again verifies the limitation of BN (degenerating when the batch size is small). At the same situation, the accuracies of HN and LN are keeping stable and insensitive to the batch size. HN converges faster than LN and can quickly reach the highest accuracy. With the increase in epoch, the vanilla CNN has the phenomenon of overfitting, which makes its test accuracies lower than those of BN and LN.

When batch size = 4, HN outperformed BN, vanilla LeNet-5, and LN in terms of convergence speed and accuracy at the beginning (Figure 6b). The test accuracies of vanilla LeNet-5, HN, LN, and BN become closer when the epoch greater than 12. Their final test accuracies differed very little.

When the batch size is 8 and 16, HN, LN, and BN converge faster than vanilla LeNet-5 (Figure 6c,d). BN slightly outperforms HN and LN in the first 5 epochs. With the increase in training epochs, their test accuracies are basically the same. Figure 6c,d show that both HN and BN can effectively improve the convergence speed of the network.

From the above analysis, we can find that BN still has the problem of accuracy degradation when the batch size is small in the FASHION-MNIST dataset. Our normalization method HN adapts to a wide range of batch sizes and dramatically improves the convergence speed and accuracy of the vanilla network.

Then, we applied these methods to the color dataset CIFAR10. Compared to the gray images, the color images have more data features. So, we additionally add GroupNorm (GN) and BatchRenorm (BRN) for comparison. Due to the simple network and limited pipeline size, we set G = 2 for GN. BRN keeps the original setting. When the training epochs size is 60, the average accuracies of the last 5 epochs are shown in Table 1.

In Table 1, we find that BN and BRN do not work well when batch size = 2. The test accuracies of HN is 6.02%, 30.24%, 0.9%, 1.84%, and 29.64% higher than that of vanilla LeNet-5, BN, GN, LN, and BRN. When batch size = 4, 8, and 16, BN has the best accuracies. HN can improve the test accuracy of vanilla LeNet-5 in all batch sizes.

The experiments on the FASHION-MNIST, and CIFAR10 datasets verify that HN can accelerate convergence and improve the test accuracy of convolutional neural network. Being independent with batch size, HN does not degenerate at smaller batch size, or saturate at larger batch size.

#### 4.2.2. VGG16

Due to the relatively simple structure of the LeNet-5, this section additionally adds a popular deep convolutional neural network VGG16. We trained, respectively, VGG16 without normalization layer (Vanilla) and VGG16 with BN or HN on the FASHION-MNIST dataset. As before, we optimized using Adam for 30 epochs, setting the initial learning rate to 0.001 and the batch sizes to 2, 4, 8, and 16. For each batch size, the curves of accuracy vs. epoch are shown in Figure 7, and the average accuracies of the last 5 epochs are shown in Table 2.

VGG16 experiments on the FASHION-MNIST dataset. When the batch size 2, 4, 8, and 16, Vanilla (VGG16 without normalization layer) experimental results are 0.100, indicating that it does not work properly in small size. If the size is larger, it can work properly, such as size = 512, etc. VGG16 with BN or HN can be trained to converge. For batch size 8 and 16, HN and BN have approximately the same effect. For batch size 2 and 4, the unstable nature of BN shows up, and HN has 10% and 5.9% better accuracy than BN, respectively.

It can be seen from the above analysis that when the size is small, VGG16 has a huge difference in the effect of adding or not adding a normalization layer to the FASHION-MNIST dataset. BN still has the problem of reduced accuracy when the batch size is small in deep convolutional neural networks. However, the HN proposed in this paper is still adaptable to all batch size cases and deep convolutional neural networks. Additionally, HN is added to the original VGG16 network as BN method, which greatly improves the convergence speed and accuracy.

#### 4.2.3. Comparisons on Residual Networks

We have analyzed the effectiveness of HN in vanilla LeNet-5 and VGG16. In this section, HN is applied to the popular ResNet-50 network to further validate the adaptability. In 2016, He et al. first proposed ResNet-50, which has 16 residual blocks containing three convolutional layers of different sizes. While comparing the effectiveness of normalization methods, we do not use training techniques, such as data augmentation and learning rate decay. The original data are read-in for network training to ensure that the comparison of different normalization methods is not affected by preprocessing.

The baseline model is ResNet-50, containing BN in its original design. The datasets used in this subsection are FASHION-MNIST and CIFAR10. In the baseline model, normalization is used after the convolution and before the ReLU. We swap it into the model in place of BN to apply HN. Adam is used as a training optimizer with a learning rate of 0.001. Let the training Epoch be 30, mini-batch sizes be 2, 4, 8, 16, and 32. In addition, we add GN, BRN for comparison. For GN, we use the recommended parameter settings in [6], where *G* = 32. For BRN, we keep the default settings in the source code.

The experimental results of the tests on the FASHION-MNIST dataset are shown in Table 3. To reducing the effect of random variation, the average test accuracies for the last five epochs are listed. The experimental results show that BN does not work well when batch size = 2. BRN converges, but has low test accuracy. HN and GN both have significant results when batch size = 2. In other batch size settings, their test accuracies are very close.

For the CIFAR10 dataset, we use the default ResNet-50 settings. Table 4 shows the average test accuracies for the last 5 epochs. The results of GN@G = 32 are not good when batch size = 2, which may be caused by the invalidity of statistics calculation. So GN@G = 2 is added for comparison too. When batch size = 2, HN achieves the highest accuracy of 72.26%, which is 0.208 better than GN(G = 2), 0.5174 higher than BRN. When the batch size = 4, the highest accuracy of HN is 0.1596 higher than BN, 0.0234 higher than GN (G = 32), and 0.0102 lower than BRN. In other batch size settings, their test accuracies are very close and stable.

The experiment results of ResNet-50 on FASHION-MNIST and CIFAR10 show that BN and BRN are batch size dependent. GN is sensitive to parameter G. HN can keep stable and high accuracy in a wide range of batch sizes.

### 4.3. Memory Requirement Analysis

In this subsection, we show the relationship between memory occupation and accuracy under vanilla, BN, and HN for LeNet-5, VGG16, and ResNet50 network models (Figure 8). The estimated total memory sizes (MB) in Figure 8 correspond to the memory requirements of the models when the batch size is 2, 4, 8, and 16. The test accuracy is the average test values of the last 5 epochs. The estimated total memory sizes are obtained by the summary function of the torchsummary in PyTorch. Due to the space limitation, we only present the experimental results on FASHION-MNIST.

The vanilla networks show minimal memory requirements because of no normalization layer. The memory requirements of BN and HN are very close, and increase with the batch size.

As can be seen from Figure 8a, in LeNet-5, when the estimated total size of HN is 0.41 MB with accuracy = 0.898, the BN achieves the same accuracy when the consuming memory reachs 0.65 MB. In Figure 8b, when the estimated total size of HN is 71.34 MB in VGG16, the BN achieves the same accuracy when the consuming memory reaches 110.93 MB. In ResNet50, HN and BN achieve accuracy 0.902 and 0.912, while the estimated total sizes of HN and BN are 99.91 MB and 172.71 MB (Figure 8c). Among the three models, HN only needs 60.1%, 64.3%, and 57.8% of the memory requirements of BN with the close accuracy.

From the above analysis, we find that models with small batch size consume less memory and are more conducive to training and applying on resource-constrained devices. Compared with BN, HN achieves higher accuracy in small batch size, so it is more suitable for resource-constrained devices.

## 5. Case Study

Brain–computer interface (BCI) constructs a communication pathway between the human brain and external devices directly without passing the muscular system. The BCI technology is widely used in assisted rehabilitation and brain-sensing games. Due to low cost and high-resolution characteristics, the electroencephalogram (EEG) signals is widely used in BCI applications. The process of EEG-BCI includes EEG signals acquisition, signal processing, and pattern recognition. Based on the advantages and features of end-to-end neural networks in pattern recognition, EEG-BCI systems gradually leaves the laboratory and is applied to portable device scenarios, such as embedded systems. As shown in Figure 9, the application of the embedded-based EEG-BCI system includes the following four steps:Training: in the laboratory experimental situation, collect enough EEG trials to train a deep neural network for patterns recognition.Deploying: deploy the pre-trained deep neural network model to the embedded device.Fine-tuning: fine-tune the deep neural network model while acquiring EEG trials.Applying: apply the fine-tuned and stabilized deep neural network model to the control of the embedded.

Wet electrode-based EEG-BCI system requires the subject to wear an EEG cap and apply conductive paste to each electrode, keeping the resistance of each electrode below 10 kΩ. However, subjects cannot guarantee that the electrode caps will be worn in the same position while migrating from the laboratory situation to the embedded device. Keeping the same resistance of each electrode is even more impossible (only guaranteed to be <10 kΩ). Based on the situation differences, the EEG trial set collected in the embedded device is not consistent with the EEG trial set used in training deep neural network, which no longer satisfy the assumption of independent identically distribution. Due to the characteristics of non-linear and non-stationary of EEG signals, the pre-trained deep neural network model needs to be fine-tuned to adapt to the embedded device situation. However, due to the storage and computational performance bottlenecks of embedded devices, we must set a limited batch size for the fine-tuning process.

In this case study, we use EEG based motor imagery BCI (MI-BCI) as an example to verify the effectiveness of HN when fine-tuning the deep neural network model. ShallowFBCSPNet (https://github.com/TNTLFreiburg/braindecode/blob/master/braindecode/models/shallow_fbcsp.py), proposed by Schirrmeister et al. in 2017, is a deep neural network designed for decoding imagined or executed tasks from raw EEG signals, which has a good performance in classifying EEG signals [31]. The BCI Competition IV dataset 2a (BCICIV 2a) is a classical EEG-based MI-BCI dataset. We take this dataset as an example to analyze and compare the performance of the HN and BN. Since BN is embedded in ShallowFBCSPNet originally, we replace BN with HN, and set max_epoch to 1600 and max_increase_epochs to 160. No extra preprocessing is performed on the EEG signals.

First, we conducted experiments in two cases, the original batch size (batch size = 60) and a smaller batch size (batch size = 8), to examine the influence of batch size on the test accuracy of HN. Table 5 shows the best prediction results in 10 epochs before the end of training with batch size = 60, while Table 6 shows the best prediction results in 10 epochs before the end of training with batch size = 8.

After replacing BN with HN, the test accuracy was improved on 6 of the 9 subjects, with a maximum improvement of 0.274 (Table 5). There was a slight decrease on the other three subjects, with a maximum reduction of 0.021. Overall, the average accuracy was improved by 0.038, which indicates that HN is more suitable for MI recognition of EEG signals when batch size = 60.

When batch size = 8, Table 6 shows that the test accuracy was improved in 8 out of 9 subjects up to 0.198 after replacing BN with HN. Overall, their average accuracy was enhanced by 0.05401. The experimental results indicate that ShallowFBCSPNet with HN is better than that of BN for MI recognition of EEG signals when the batch size is small.

To demonstrate the real application scenario of the deep learning-based EEG signals application system, the experiments were conducted with EEG signals from subject A as training (batch size = 8), followed by fine-tuning (batch size = 2) and testing on subject B as the user. We train the model with the EEG signals from subjects 2, 4, 6, 8, and 1, and fine-tune and test the model on subjects 1, 3, 5, 7, and 9 as users. During fine-tuning, we randomly selected 20% of the EEG signals from subject B. Finally, the best prediction result of the last 10 epochs of the fine-tuned model is shown in Table 7.

As shown in Table 7, when the subject and the user are not the same subject, the model fine-tunes with a smaller batch size. Although there is fewer fine-tuning data, the accuracy of ShallowFBCSPNet decreases greatly, which indicates that the embedded device applications of deep learning-based EEG signal recognition model still have a long way to go. Comparing with HN and BN, HN demonstrates a better accuracy in five pairs of experiments, while BN does not show enough advantages. Overall, the average accuracy of the HN is 4.4% higher than that of BN, which indicates that HN is more suitable for deep neural network recognition model on resource-constrained devices.

## 6. Conclusions

Habituation is a simple memory that changes over time and inhibits neural responses to repetitive, neutral stimuli. This adaptive mechanism allows organisms to focus their attention on the most salient signals in the environment. The Drosophila olfactory system is based on a “negative image” model of habituation that filters redundant information and enhances olfactory stability. Inspired by the circular mechanism of the Drosophila olfactory system, we propose a novel normalization method, habituation normalization (HN), with three characteristics of habituation in biology: stimulus adaptation, stimulus specificity, and reversibility. HN first eliminates the “negative image” obtained by habituation and then calculates the overall statistics to achieve normalization.

We apply HN to LeNet-5, VGG16, and ResNet-50. Experiments on three benchmark datasets show that HN can effectively accelerate network training and improve the test accuracy. By comparing with other normalization methods (LN, BN, GN, and BRN), the experimental results verify that HN can be used in a wide range of batch sizes, and show good robustness. Finally, we apply HN to a deep learning-based EEG signal application system. Experiment results in two cases (train on A, test on A; train on A, trial on B) show that HN are more suitable for deep learning network applications in resource-constrained devices.

As future work, we will extend HN to other types of deep learning networks, such as recurrent neural network (RNN/LSTM) or Generative Adversarial Network (GAN).

## Figures and Tables

**Figure 1 sensors-22-09940-f001:**
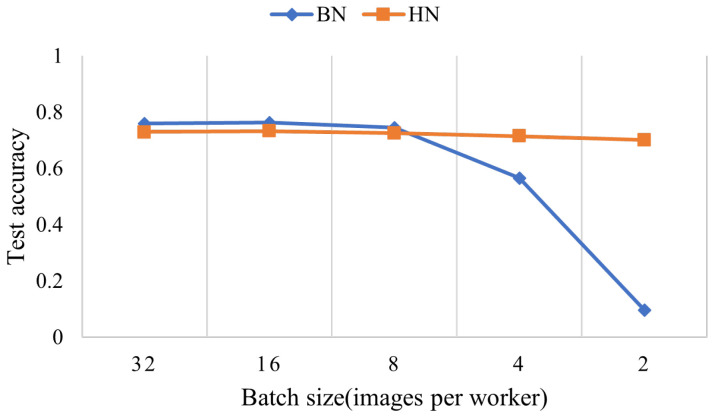
CIFAR10 classification test accuracy vs. batch sizes. This is a ResNet-50 model trained in the CIFAR10, which shows that small batches may lead to lower accuracy dramatically.

**Figure 2 sensors-22-09940-f002:**
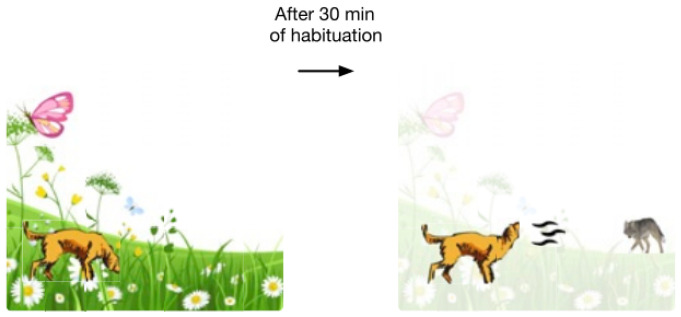
When a dog sits in the garden and get used to the smell of flowers, the dog can perceive any changes in the environment (for example, a coyote that appears in the distance) [9]. In this scene, the dog’s smell of flowers gradually fades away, when the dog is habitual. Then the new coming smells will be magnified and be detected easily.

**Figure 3 sensors-22-09940-f003:**
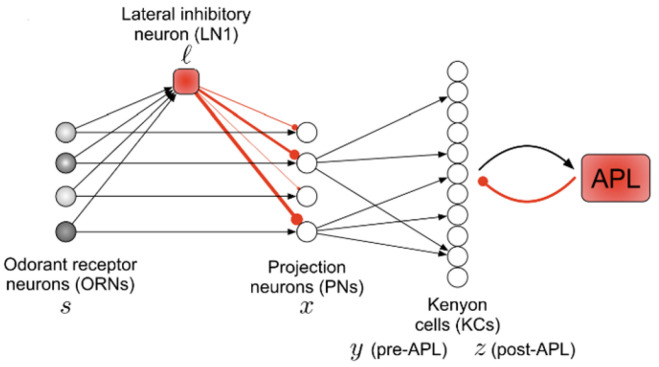
Overview of the fruit fly olfaction circuit. For each input odor, ORNs each fire at some rate. ORNs send feed-forward excitation to PNs of the same type. A single lateral inhibitory neuron (LN1) also receives feed-forward inputs from ORNs and then sends inhibition to PNs. The locus of habituation lies at the LN1 → PN synapse, and the weights of these synapses are shown via red line thickness. PNs then project to KCs, and each KC samples sparsely from a few PNs [9].

**Figure 4 sensors-22-09940-f004:**
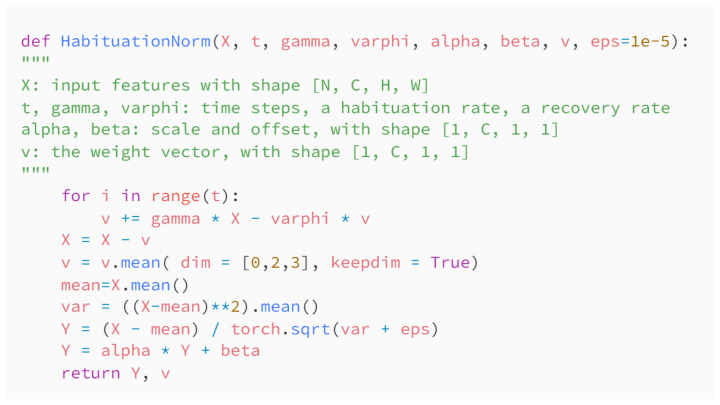
Python code of habituation normalization base on Pytorch.

**Figure 5 sensors-22-09940-f005:**
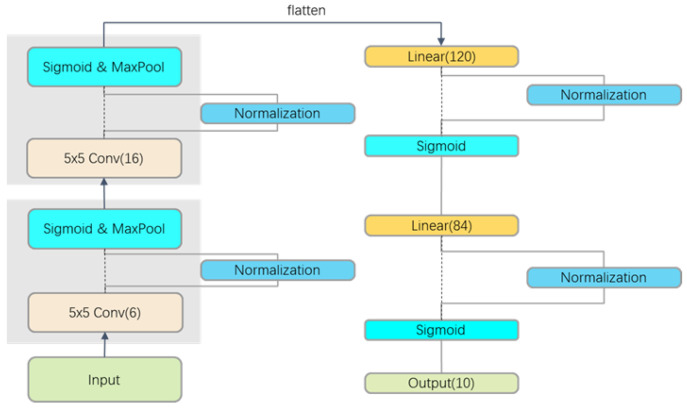
Convolutional Neural Network based on LeNet-5.

**Figure 6 sensors-22-09940-f006:**
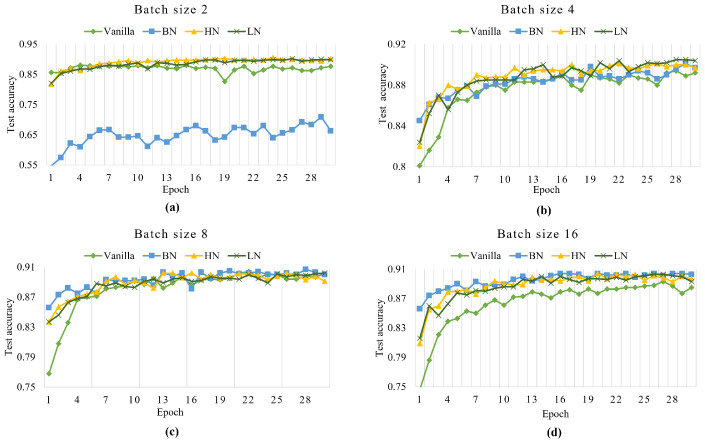
The curves of training epochs vs. test accuracy on dataset FASHION-MNIST with (**a**) vanilla LeNet-5, (**b**) BN, (**c**) HN, and (**d**) LN.

**Figure 7 sensors-22-09940-f007:**
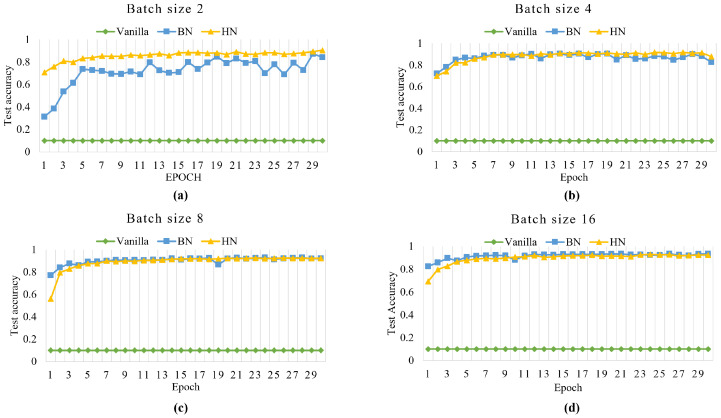
The curves of training epochs vs. test accuracy on dataset FASHION-MNIST with VGG16. (**a**) batch size = 2. (**b**) batch size = 2. (**c**) batch size = 2. (**d**) batch size = 2.

**Figure 8 sensors-22-09940-f008:**
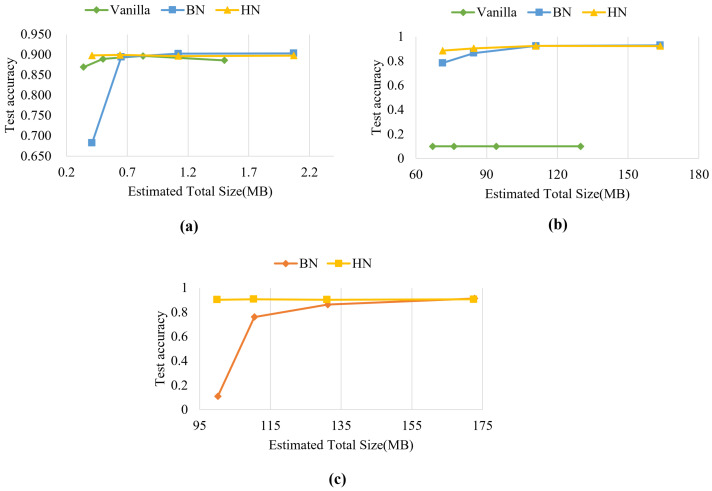
The curves of estimated total size (MB) vs. test accuracy in different normalization layer corresponding to batch size on dataset FASHION-MNIST. (**a**) LeNet-5 @epoch=30. (**b**) VGG16 @epoch = 30. (**c**) ResNet50 @epoch = 30.

**Figure 9 sensors-22-09940-f009:**
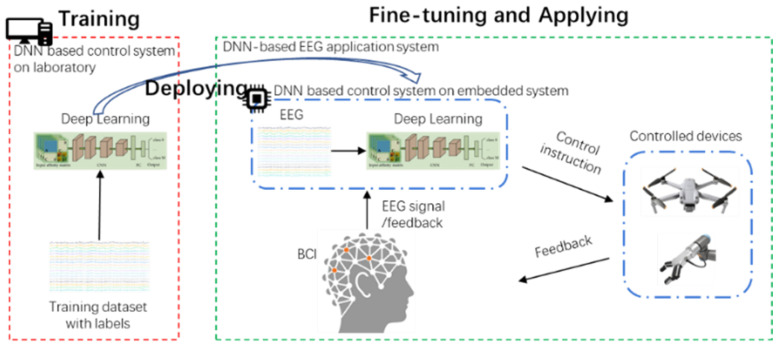
EEG signal application system based on deep learning and its application process.

**Table 1 sensors-22-09940-t001:** Experimenting in CIFAR10. The average test accuracy of the last 5 epochs on vanilla LeNet-5, BN, GN, LN, BRN, and HN.

Batch Size	2	4	8	16
Vanilla	0.5408	0.5496	0.5702	0.5796
BN	0.2986	**0.673**	**0.6854**	**0.6564**
GN	0.592	0.618	0.6056	0.6096
LN	0.5826	0.5894	0.6178	0.598
BRN	0.3046	0.6328	0.6476	0.6498
HN	**0.601**	0.6114	0.6232	0.6278

**Table 2 sensors-22-09940-t002:** Experimenting in FASHION-MNIST, the average test accuracy of the last 5 epochs on vanilla VGG16, BN, and HN.

Batch Size	2	4	8	16
Vanilla	0.100	0.100	0.100	0.100
BN	0.784	0.865	0.923	**0.929**
HN	**0.884**	**0.924**	**0.924**	0.923

**Table 3 sensors-22-09940-t003:** Experimenting in the FASHION-MNIST dataset with ResNet-50, The average test accuracy of the last 5 epochs on ResNet-50 (BN), BRN GN@G = 32, and HN.

Batch Size	2	4	8	16	32
BN	0.109	0.7606	0.8618	**0.9124**	**0.9176**
BRN	0.479	0.904	**0.9132**	0.9108	0.9138
GN@G = 32	**0.9062**	**0.9064**	0.9038	0.9066	0.9046
HN	0.9024	0.906	0.9018	0.9056	0.9074

**Table 4 sensors-22-09940-t004:** Experimenting in the CIFAR-10 dataset with ResNet-50, The average test accuracy of the last 5 epochs on ResNet-50 (BN), BRN, GN@G = 32, GN@G = 2, and HN.

Batch Size	2	4	8	16	32
BN	0.0954	0.5658	0.7354	0.7344	0.7504
BRN	0.2052	**0.7356**	**0.7370**	**0.7626**	**0.7552**
GN@G = 32	0.1	0.702	0.7016	0.7138	0.714
GN@G = 2	0.7018	0.701	0.7186	0.721	0.7224
HN	**0.7226**	0.7254	0.7326	0.7218	0.7264

**Table 5 sensors-22-09940-t005:** Test accuracies of EEG signal classification using ShallowFBCSPNet (batch size = 60) on the BCIC IV 2a dataset. Δ is the improvement of test accuracy by HN.

Accuracy	1	2	3	4	5	6	7	8	9	Average
BN	0.840	**0.483**	0.882	**0.740**	0.288	0.535	**0.924**	0.778	0.764	0.693
HN	**0.865**	0.472	**0.896**	0.719	**0.563**	**0.569**	0.910	**0.799**	**0.788**	**0.731**
Δ	**0.025**	−0.011	**0.014**	−0.021	**0.275**	**0.034**	−0.014	**0.021**	**0.024**	**0.038**

**Table 6 sensors-22-09940-t006:** Test accuracies of EEG signals classification using ShallowFBCSPNet (batch size = 8) on the BCIC IV 2a dataset. Δ is the improvement of test accuracy by HN.

Accuracy	1	2	3	4	5	6	7	8	9	Average
BN	0.792	0.431	0.837	0.642	0.368	0.517	**0.927**	0.740	0.736	0.666
HN	**0.840**	**0.465**	**0.865**	**0.708**	**0.566**	**0.535**	0.906	**0.785**	**0.806**	**0.720**
Δ	**0.048**	**0.034**	**0.028**	**0.066**	**0.198**	**0.018**	−0.021	**0.045**	**0.070**	**0.046**

**Table 7 sensors-22-09940-t007:** Accuracies obtained under different training and test users. Training model with EEG from subject i (batch size = 8), fine-tuning with 20% EEG from subject j (batch size = 2), and then test on subject j. i→j represents training on subject i, testing on subject j. Δ is the improvement of test accuracy by HN.

Accuracy	2→1	4→3	6→5	8→7	1→9	Average
BN	0.507	0.628	0.288	0.566	0.587	0.515
HN	**0.594**	**0.635**	**0.340**	**0.597**	**0.628**	**0.559**
Δ	**0.087**	**0.007**	**0.052**	**0.031**	**0.041**	**0.044**

## Data Availability

Not applicable.

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
