# Peer review of "Improving Network Training on Resource-Constrained Devices via Habituation Normalization†"

_sensors, 2022, doi:10.3390/s22249940_

Round 1

Reviewer 1 Report

- Arrange keywords alphabetically

- Tables 5 & 6 's widths need to be adjusted

- Thorough revision for the english language is required for the final submission 

Author Response

 Thanks for your valuable comments. We have revised the manuscript. All revisions have been marked up using “Track Changes”.

Comment 1: Arrange keywords alphabetically.

Response 1: We have arranged the keywords alphabetically.

Comment 2: Tables 5 & 6 's widths need to be adjusted.

Response 2: We have adjusted the widths of Table 5 and Table 6.

Comment 3: Thorough revision for the English language is required for the final submission.

Response 3: We have revised some expressions, checked the grammar, words, punctuation and rewrote some sentences. The traces of modification edited by the authors are retained in the revised manuscript.

Reviewer 2 Report

Dear Authors,

Please find my suggestions below.

This is very promising study which is dealing with improving network training  by application of novel method habituation normalization and possible application in some real situations.

Make two sentences from the sentence which starts in line 2 and ends in line 4.

The sentence which starts in line 19 and ends in line 22 is too long. Make two sentences.

Make two sentences from the sentence which starts in line 29 and ends in line 31.

Make two sentences from the sentence which starts in line 57 and ends in line 59.

The sentence which starts in line 66 and ends in line 69 is too long. Make two sentences.

The sentence which starts in line 84 and ends in line 86 is too long. Make two sentences.

Make two sentences from the sentence which starts in line 126 and ends in line 128.

Make two sentences from the sentence which starts in line 128 and ends in line 131.

The sentence which starts in line 154 and ends in line 157 is too long. Make two sentences.

Figures are missing in document.

Kindly consider to highlight in document all changes of text.

In line 324 number of table is missing.

Wish you all the best in future work.

Author Response

Thanks for your valuable comments. We have revised the manuscript. All revisions have been marked up using “Track Changes”.

Comment 1: Make two sentences from the sentence which starts in line 2 and ends in line 4.

       The sentence which starts in line 19 and ends in line 22 is too long. Make two sentences.

Make two sentences from the sentence which starts in line 29 and ends in line 31.

Make two sentences from the sentence which starts in line 57 and ends in line 59.

The sentence which starts in line 66 and ends in line 69 is too long. Make two sentences.

The sentence which starts in line 84 and ends in line 86 is too long. Make two sentences.

Make two sentences from the sentence which starts in line 126 and ends in line 128.

Make two sentences from the sentence which starts in line 128 and ends in line 131.

The sentence which starts in line 154 and ends in line 157 is too long. Make two sentences.

Response 1: Thanks for your kindly suggestions. We have rewritten these sentences with short ones. Also, other grammars, expressions are also been revised. The line number of the rewritten sentence are list below.

The line number of the original sentence.        The line number of the revised one.

starting in line 2 and ending in line 4               Replace by two sentences: lines 2-4.

starting in line 19 and ending in line 22           Replace by two sentences: lines 20-25.

starting in line 29 and ending in line 31           Replace by two sentences: lines 32-35.

starting in line 57 and ending in line 59           Replace by two sentences: lines 63-65.

starting in line 66 and ending in line 69           Replace by two sentences: lines 72-74.

starting in line 84 and ending in line 86           Replace by two sentences: lines 92-94.

starting in line 126 and ending in line 128        Rewrite the sentence: lines 139-141.

starting in line 128 and ending in line 131        Rewrite the sentence: lines 141-145.

starting in line 154 and ending in line 157        Rewrite the sentence: lines 169-173

Comment 2: Figures are missing in document.

Response 2: They are fixed now. All figures can been view properly.

Comment 3: Kindly consider to highlight in document all changes of text.

Response 3: Thanks for you kindly reminder. The traces of modification edited by the authors are retained in the revised manuscript.

Comment 4: In line 324 number of the table is missing.

Response 4: Sorry for my careless. The number of the table has been added.

Reviewer 3 Report

Manuscript ID: sensors-1944386 

Title: Improving Network Training on Resource-Constrained Devices via Habituation-based Normalization.

I read the manuscript carefully and have some suggestions which are given below:

1.      The authors should pay attention to contributions that are unclear.

2.      In Figure 2, the habitation concept is not clear; the authors should have some more comments on it.

3.      In order to understand the hierarchy of this research, authors should add a block diagram.

4.      What is the relation between equation (4) and equation (6)?

5.      In the experimental section, which kind of hardware has been used?

Author Response

Thanks for your valuable comments. We have revised the manuscript. All revisions have been marked up using “Track Changes”.

Comment 1: The authors should pay attention to contributions that are unclear.

Response 1: Thanks for your reminder. We have rewritten some items in the contributions.

Comment 2: In Figure 2, the habitation concept is not clear; the authors should have some more comments on it.

Response 2: Habituation is not a concept in the computer field, but a concept in the biological field. In this manuscript, we use the word “habituation” to express the inspiration of this method. In paragraph 1, subsection 2.2, “habitation” is defined and explained.

       In Figure 2, the dog's smell of flowers gradually fades away, when the dog is habitual. Then the new coming smells will be magnified and be detected easily. We also add comments to the caption of Figure 2 for further explanation.

Comment 3: In order to understand the hierarchy of this research, authors should add a block diagram.

Response 3: In neural networks, the normalization function that is inserted before the activation function can be regarded as a normalization layer. It is part of a neural network. For example, in Figure 5, five normalization functions are inserted before 5 activation functions respectively.

We have shown the block diagram of LeNet-5 with normalization in Figure 5. The insertion position of HN is relatively certain before the activation functions. Therefore, for VGG16 and ResNet, we do not repeat the block diagrams.

Comment 4: What is the relation between equation (4) and equation (6)?

Response 4: Equation (4) is the expression for GN. equations (5) and (6) are the equations for  and  in equation (4). We have replaced  and  with  and  in equation (4).

Comment 5: In the experimental section, which kind of hardware has been used?

Response 5: The experiments are performed on a machine with Intel(R) Xeon(R) Silver 4114 CPU @2.20GHz 2.19GHz (2 processors), 128GB of RAM, NVIDIA Quadro P5000 graphics card, using windows server 2016. We also add the hardware description into subsection 4.1.

In the case study, we performed the experiments on an embedded system with Broadcom BCM2836 900MHz, 1GB SDRAM @400MHz.

Reviewer 4 Report

I thank the authors and moving to the paper itself. As far as authors are presenting:

a. Habituation normalization (HN) can speed up neural network training and improve the model accuracy

b. HN is suitable for fine-tuning and applications for limited computing power and memory

I am sorry but I have a number of problems with this paper, broadly:

1.     What is the nobility of this present study except introducing new technique or any other aspect?

2.     Author used Habituation normalization for modelling, I suggest please add any hybrid method to satisfy your model efficacy (Ex. Of Hybrid method: MLR-GWO, ANFIS-PSO, SVM-HHO).

3.     So many constraints like RMSE, R2, AUC, WI, NSE, MSE, PBIAS are used for evaluation purposes. I recommend author must use four evaluating constraint for better analysis.

4.     What is the limitation of your study?

5.     Provide a table for different combination of used parameter towards model development

6.     Compare your research with other study for performance purposes

7.     Future scope must be added at last portion of reference section.

8.     Detail description must be provided for figure 8 and 9

9.     Histogram/box plot, scatter plot, comparison plot must be added for better accuracy of model

1.  Add more recent references (2021 and 2022) in context to this research

1.  All citation must added in the reference section and vice versa

1.  Give abbreviation when it appears first.

1.  Check the format of reference as per journal guideline

1.  Pl. read the paper carefully as some sentences are incomplete and there are some missing article and verbs in the sentences.

1.  The English is so bad that it is very difficult to understand while going through the text. The authors may be asked to rectify the mistakes by take up the help of a person who had English as first language.

1.  Please change Fig 1 to 9. As it is not visible.

1.  Author must provide details of data collected/experimental one for research purposes.

Author Response

Thanks for your valuable comments. We have revised the manuscript. All revisions have been marked up using “Track Changes”.

Comment 1: What is the nobility of this present study except introducing new technique or any other aspect?

Response 1: Inspired by the habituations in the biological field, we proposed a normalization method, Habituation Normalization (HN), which can be applied to neural networks. HN can accelerate the convergence and accuracy of networks in a wide range of batch sizes. The experiments on LeNet-5, VGG16 and ResNet-50 show that HN has good adaptability. At last, the application of HN to deep learning-based EEG signal application system shows that HN is suitable for deep neural networks running on resource-constrained devices.

       More details can be seen in our contributions in section 1.

Comment 2: Author used Habituation normalization for modelling, I suggest please add any hybrid method to satisfy your model efficacy (Ex. Of Hybrid method: MLR-GWO, ANFIS-PSO, SVM-HHO).

Response 2: Indeed, Multiple Linear Regression (MLR)+Grey Wolf Optimization algorithm (GWO), Adaptive Neuro-Fuzzy Inference System(ANFIS) + Particle Swarm Optimization (PSO), Support Vector Machine (SVM) + Harris Hawks Optimization (SVM-HHO) are some hybrid methods used for classification. But I don’t think they can be added into HN as part of neural network.

       Normalizing hidden features in neural networks can speed up the network training. It is part of the neural networks, and cannot work alone.

Comment 3: So many constraints like RMSE, R2, AUC, WI, NSE, MSE, PBIAS are used for evaluation purposes. I recommend author must use four evaluating constraint for better analysis.

Response 3: The habituation normalization is part of neural networks. It cannot run alone as a classification model. We reviewed the papers about normalization once more [1-5]. They all do not evaluate the normalization method with index like RMSE, R2, AUC, WI, NSE, MSE, PBIAS. Most importantly, the classification results of our network output are determined, not probabilities.

Comment 4: What is the limitation of your study?

Response 4: Yes, our normalization methods has limitations. First of all, while training with a larger batch size, our method is not the best one. The limitation has been described in the experiment. Secondly, we just apply HN on the convolutional neural network. We have not yet verified the method on other deep learning models, such as recurrent neural network (RNN/LSTM) or Generative Adversarial Network (GAN). Applying HN to RNN and GAN is our future work.

Comment 5: Provide a table for different combination of used parameter towards model development

Response 5: We set  as the default setting. The parameters are determined through multiple experiments and used in all experiments. We have declared it in subsection 4.1

Comment 6: Compare your research with other study for performance purposes

Response 6: In section 4, we have compared HN with other normalization methods, such as BN, GN, LN and BRN, under different deep learning model, such as LeNet-5, VGG16 and ResNet. The results show that HN have best performance when the batch-size is small. Also, it can accelerate the convergence and accuracy of networks in wide range of batch sizes.

Comment 7: Future scope must be added at last portion of reference section.

Response 5: We have pointed out the future works in section 6. As future work, we will extend HN to other types of deep learning networks, such as recurrent neural network (RNN/LSTM) or Generative Adversarial Network (GAN).

Comment 8: Detail description must be provided for figure 8 and 9.

Response 8: Detail description for Figure 8 and 9 can be found in subsection 4.3 and section 5 respectively.

Comment 9: Histogram/box plot, scatter plot, comparison plot must be added for better accuracy of model.

Response 9: The graphs and tables have been used to show the results, such as Table 1-7, Figure 6-8. There are already 9 figures, so we show some of the results by listing the data in the tables.

Comment 10: Add more recent references (2021 and 2022) in context to this research.

Response 10: We have added more references, which are also cited in the main body of the manuscript, such as reference [2, 7, 14, 15]. These papers are published in 2021 and 2022.

Comment 11: All citation must be added in the reference section and vice versa

Response 11: We have checked the references and make sure all references are cited in the manuscript.

Comment 12: Give abbreviation when it appears first.

Response 12: We have checked the paper about the abbreviations.

Comment 13: Check the format of reference as per journal guideline.

Response 13: Thanks for your reminding, we have checked the format of references.

Comment 14: Pl. read the paper carefully as some sentences are incomplete and there are some missing article and verbs in the sentences.

Response 14: We have revised some expressions, checked the grammar, words, punctuation and rewrote some sentences. The traces of modification edited by the authors are retained in the revised manuscript.

Comment 15: The English is so bad that it is very difficult to understand while going through the text. The authors may be asked to rectify the mistakes by take up the help of a person who had English as first language

Response 15: We have revised some expressions, checked the grammar, words, punctuation and rewrote some sentences. The traces of modification edited by the authors are retained in the revised manuscript.

Comment 16: Please change Fig 1 to 9. As it is not visible.

Response 16: We have set the figures correctly. The figures are visible now.

Comment 17: Author must provide details of data collected/experimental one for research purposes.

Response 17: All data used in the experiments are public dataset. They can be downloaded from the Internet.

Round 2

Reviewer 2 Report

Dear Authors,

Thank you very much for revised version of manuscript. Manuscript is now significantly improved especially when figures are added. You have not make two sentences from one in three sections (139-141, 141-145, 169-173).

Wish you all the best in future work.

Reviewer 3 Report

It can be accepted in its present form.